# Ferrocene-Based Terpolyamides and Their PDMS-Containing Block Copolymers: Synthesis and Physical Properties

**DOI:** 10.3390/polym14235087

**Published:** 2022-11-23

**Authors:** Irrum Mushtaq, Erum Jabeen, Zareen Akhter, Fatima Javed, Azfar Hassan, Muhammad Saif Ullah Khan, Faheem Ullah, Faiz Ullah Shah

**Affiliations:** 1Department of Biological Sciences, National University of Medical Sciences, The Mall Road, Rawalpindi 46000, Pakistan; 2Department of Chemistry, Quaid-i-Azam University, Islamabad 45320, Pakistan; 3Department of Chemistry, Faculty of Science, Allama Iqbal Open University, Islamabad 44310, Pakistan; 4Department of Chemistry, Shaheed Benazir Bhutto Women University, Peshawar 25000, Pakistan; 5Department of Chemical and Petroleum Engineering, University of Calgary, Calgary, AB T2N 1N4, Canada; 6School of Materials and Mineral Resources Engineering, Engineering Campus, University Sains Malaysia, Nibong Tebal 14300, Malaysia; 7Chemistry of Interfaces, Luleå University of Technology, SE-971 87 Luleå, Sweden

**Keywords:** terpolymers, copolymers, thermal stability, surface morphology, water absorption behaviour, DFT studies

## Abstract

Aromatic polyamides are well-known as high-performance materials due to their outstanding properties making them useful in a wide range of applications. However, their limited solubility in common organic solvents restricts their processability and becomes a hurdle in their applicability. This study is focused on the synthesis of processable ferrocene-based terpolyamides and their polydimethylsiloxane (PDMS)-containing block copolymers, using low-temperature solution polycondensation methodology. All the synthesized materials were structurally characterized using FTIR and ^1^H NMR spectroscopic techniques. The ferrocene-based terpolymers and block copolymers were soluble in common organic solvents, while the organic analogs were found only soluble in sulfuric acid. WXRD analysis showed the amorphous nature of the materials, while the SEM analysis exposed the modified surface of the ferrocene-based block copolymers. The structure–property relationship of the materials was further elucidated by their water absorption and thermal behavior. These materials showed low to no water absorption along with their high limiting oxygen index (LOI) values depicting their good flame-retardant behavior. DFT studies also supported the role of various monomers in the polycondensation reaction where the electron pair donation from HOMO of diamine monomer to the LUMO of acyl chloride was predicted, along with the calculation of various other parameters of the representative terpolymers and block copolymers.

## 1. Introduction

The design and synthesis of new multifunctional materials are always the key aspects of their development and wide range of applications [1,2,3]. Aromatic polyamides are the class of high-performance polymers possessing high thermal and oxidative stability, exceptional wear and abrasion resistance, and outstanding mechanical and dielectric properties [2,3,4,5]. Aromatic polyamides have been synthesized for various potential applications such as heat protective clothing, rope, gears, medical implants, etc. [2,6,7]. The major obstacles in the use of aromatic polyamides are their limited solubility in common organic solvents and the high heat resistance that makes their processing difficult [8,9]. Many studies have shown strategies to improve the processability of polyamides via the structural design of new monomers [8,9,10,11]. Aromatic diamine is one of the major building blocks to tailor the properties of the synthesized polyamides [12,13]. Various aromatic diamines possessing flexible spacers such as ether, methylene, and sulfone were incorporated into the polymer backbone to improve the flexibility of the polymer chains and hence their processability, along with maintaining their structure–property relationship [12,14].

The organometallic polyamides possessing a metal center in the polymer backbone are under consideration for the unique combination of properties offered by inorganic and organic centers in the polymer backbone [15]. Ferrocene is an organometallic compound that has been widely used in many approaches to utilize its good redox behavior, thermal, photo, air, water, and oxidative stability [16,17,18]. Due to the excellent properties of ferrocene moiety, it has been incorporated in the main chain or as a pendant group in the polymer backbone for a variety of applications such as electrical devices, photo-resistive semiconductors, ion sensing, energy storage devices, thermal and moisture resistant coatings, flame-retardant materials, etc. [19,20,21,22,23]. A combination of ferrocene moiety with flexible linkages in the backbone of polyamides has been widely used as a strategy to improve their processability along with their improved physiochemical properties [17,18].

Polyamides have been made processable via terpolymerization or copolymerization processes in several approaches [17]. A terpolymer is a polymer comprising three different monomers, incorporating the properties of each monomer, while a copolymer consists of more than one repeating unit of different types of monomers in the polymer chain. Polydimethylsiloxane (PDMS) is a siloxane-containing polymer that has been incorporated into different types of block copolymers to integrate its interesting properties such as thermal stability, moisture resistance, and flexibility—hence the processability of the desired product [24]. PDMS-containing polyamides have been mostly synthesized through solution polycondensation of the rigid polymer block and flexible PDMS-containing block for applications such as implants, contact lenses, gas separation membranes, photo-resistive coatings, and emulsifiers [25,26,27,28].

In the present study, we have attempted to keep in view the importance of terpolymerization and copolymerization as promising approaches to synthesizing processable aromatic polyamides. The low-temperature solution polycondensation method was used to synthesize the polymeric material. To investigate the structure–property relationship, various combinations of aromatic diamines and diacid chlorides were used keeping the ratios of each monomer constant. In the first series, ferrocene-based terpolyamides were synthesized, while in the second series block copolymers of the ferrocene-based terpolyamides containing amino-terminated polydimethylsiloxane (PDMS) were synthesized. In the third and fourth series, corresponding organic analogs of ferrocene-containing terpolyamides and their PDMS-containing block copolymers were synthesized, respectively, using two different diacid chloride monomers, i.e., terephthaloyl chloride and isophthaloyl chloride. The novelty of this work resides in using different combinations of aromatic diamines which have flexible ether, sulfone, and methylene linkages along with kinked ferrocene and PDMS moieties that have never been explored before for ferrocene-based aromatic polyamides, as well as their PDMS-containing block copolymers. We aimed to tailor the structure–property relationship of the synthesized material using this strategy.

## 2. Experimental Materials

Ferrocene, acetyl chloride, aluminum chloride, and 4,4′-diaminodiphenyl sulfone were purchased from Fluka, Switzerland, and used without further purification. Sodium hypochlorite solution (5%), anhydrous sodium carbonate, and thionyl chloride were obtained from Merck, Germany. The following items of 4,4′-Diaminodiphenyl ether, 4,4′-diaminodiphenyl methane, bis(3-aminopropyl) terminated polydimethylsiloxane (PDMS, y = 30), terephthaloyl chloride, and isophthaloyl chloride were purchased from Aldrich, Germany. Tetrahydrofuran (THF), dichloromethane, diethyl ether, triethylamine, N,N′-dimethylsulfoxide (DMSO), methanol, n-hexane, toluene, 1, 4-dioxane, ethanol, chloroform, N,N′-dimethylformamide (DMF), m-cresol, and N-methyl-2-pyrrolidone were obtained from Riedel-de Haën, Germany. All the chemicals used were of the highest purity and the solvents were distilled and freshly dried before use [18]. The reactions were carried out under an inert nitrogen atmosphere. The pre-coated Kieselgel 60HF TLC plates were used for thin-layer chromatography to monitor the progress of the reaction and the purity of the product.

### 2.1. Synthesis

1,1′-ferrocenedicarbonyl chloride: The monomer 1,1′-ferrocenedicarbonyl chloride was prepared in three steps (Figure 1). The first step was the synthesis of 1,1′-diacetyl ferrocene [29] followed by the preparation of 1,1′-ferrocene dicarboxylic acid [30] in the second step according to the reported literature. In the third step, 1,1′-ferrocenedicarbonyl chloride was prepared by an improved and efficient method as previously reported by our group [18].

Ferrocene-based aromatic terpolyamides and their organic analogs: In a prebaked two-necked round bottom flask equipped with a condenser and magnetic stirrer, different combinations of diamines 50:50 (0.92 mmol each) were dissolved in 20 mL of freshly dried THF and treated with an excess of triethylamine (10 mL) under a nitrogen atmosphere. Then, 1, 1′-ferrocene dicarboxylic acid chloride (0.6 g, 1.92 mmol) solution in THF was added dropwise at 0 °C with vigorous stirring over half an hour. Next, the temperature was raised slowly to room temperature, and the reaction mixture was stirred for 5 h and then refluxed for 1 h. The precipitates obtained on cooling were filtered and washed with methanol and THF several times. A dark brown polymer was obtained which was dried under a vacuum for 24 h. The organic analogs were synthesized by a similar method using terephthaloyl chloride and isophthaloyl chloride monomers. The scheme for the synthesis of terpolyamides is given in Figure 2.

F1: 4,4′-Diaminodiphenyl sulfone 0.2 g (0.92 mmol), 4,4′-diaminodiphenyl ether 0.18 g (0.92 mmol), and 1,1′-ferrocene dicarbonyl chloride 0.6 g (1.92 mmol). Dark brown solid mass. Yield: 58%; FTIR in υ (cm^−1^): 3342 (m, N-H stretch), 3072, 3033 (w, C-H stretch Ar), 2976, 2936 (w, C-H stretch Al), 1634 (s, amide I), 1590 (s, amide II), 1315, 1140 (s, SO_2_), 1241 (s, COC), 813 (m, C_6_H_4_ bend), 488 (m, Fe-Cp); and ^1^H NMR (300 MHz, DMSO-d6, δ ppm): 4.97–4.45 (br, 8H, Fe-Cp), 7.59–6.08 (br, 16H, Ar).

F2: 4,4′-Diaminodiphenyl methane 0.18 g (0.92 mmol), 4,4′-diaminodiphenyl ether 0.18 g (0.92 mmol), and 1,1′-ferrocene dicarbonyl chloride 0.6 g (1.92 mmol). Dark brown solid mass. Yield: 68%; FTIR in υ (cm^−1^): 3315 (m, N-H stretch), 3132, 3051 (w, C-H stretch Ar), 2928, 2866 (w, C-H stretch Al), 1643 (s, amide I), 1615 (s, amide II), 1237 (s, COC), 823 (m, C_6_H_4_ bend), 497 (m, Fe-Cp); and ^1^H NMR (300 MHz, DMSO-d6, δ ppm): 4.88–4.51 (m, 8H, Cp), 7.69–6.47 (m, 16H, Ar), 3.36 (s, 2H, Ar-CH_2_).

F3: 4,4′-Diaminodiphenyl ether 0.18 g (0.92 mmol), ethylene diamine 0.06 g (0.92 mmol), and 1,1′-ferrocene dicarbonyl chloride 0.6 g (1.92 mmol). Light brown solid mass. Yield: 65%; FTIR in υ (cm^−1^): 3271 (m, N-H stretch), 3049, 3016 (w, C-H stretch Ar), 2979–2944 (w, C-H stretch Al), 1622 (s, amide I), 1598 (s, amide II), 1216 (s, COC), 800 (m, C_6_H_4_ bend), 483 (m, Fe-Cp); and ^1^H NMR (300 MHz, DMSO-d6, δ ppm): 4.84–4.40 (br, 8H, Cp), 7.65–6.97 (m, 16H, Ar), 1.18 (t, 4H, CH_2_-Al).

T1: 4,4′-Diaminodiphenyl sulfone 0.2 g (0.92 mmol), 4,4′-diaminodiphenyl ether 0.18 g (0.92 mmol), and terephthaloyl chloride 0.4 g (1.92 mmol). Off-white solid mass. Yield: 97%; FTIR in υ (cm^−1^): 3293 m (N-H stretch), 3125, 3043 (w, C-H stretch Ar), 2976, 2945 (w, C-H stretch Al), 1644 (s amide I), 1593 (s, amide II), 1319, 1146 (s, SO_2_), 1217 (s, COC), 828 (m, C_6_H_4_ bend); and ^1^H NMR (300 MHz, DMSO-d6, δ ppm): 7.87–6.61 (mult. 12H, Ar).

T2: 4,4′-Diaminodiphenyl methane 0.18 g (0.92 mmol), 4,4′-diaminodiphenyl ether 0.18 g (0.92 mmol), and terephthaloyl chloride 0.4 g (1.92 mmol). Off-white solid mass. Yield: 95%; FTIR in υ (cm^−1^): 3284 (m, N-H stretch), 3138, 3067 (w, C-H stretch Ar), 2928, 2854 (w, C-H stretch Al), 1649 (s, amide I), 1610 (s, amide II), 1239 (s, COC), 822 (m, C_6_H_4_ bend); and ^1^H NMR (300 MHz, H_2_SO_4_-d2, δ ppm): 7.79–6.68 (mult. 20H, Ar), 2.21 (sing. 2H, Ar-CH_2_).

T3: 4,4′-Diaminodiphenyl ether 0.18 g (0.92 mmol), ethylene diamine 0.06 g (0.92 mmol), terephthaloyl chloride 0.4 g (1.92 mmol). Yellow solid mass. Yield: 85%; FTIR in υ (cm^−1^): 3294 (m, N-H stretch), 3076, 3045 (w, C-H stretch Ar), 2979, 2945 (w, C-H stretch Al), 1634 (s, amide I), 1590 (s, amide II), 1216 (s, COC), 807 (m, C_6_H_4_ bend); and ^1^H NMR (300 MHz, H_2_SO_4_-d2, δ ppm): 7.73–6.68 (mult. 16H, Ar), 3.56 (mult. 4H, CH_2_-Al).

I1: 4,4′-Diaminodiphenyl sulfone 0.23 g (0.92 mmol), 4,4′-diaminodiphenyl ether 0.18 g (0.92 mmol), and isophthaloyl chloride 0.4 g (1.92 mmol). Off-white solid mass. Yield: 89%; FTIR in υ (cm^−1^): 3272 (m, N-H stretch), 3072, 3031 (w, C-H stretch Ar), 2976, 2936 (w, C-H stretch Al), 1652 (s, amide I), 1592 (s, amide II), 1328, 1148 (s, SO_2_), 1217 (s, COC), 815 (m, C_6_H_4_ bend); and ^1^H NMR (300 MHz, H_2_SO_4_-d2, δ ppm): 7.90–7.09 (mult. 12H, Ar).

I2: 4,4′-Diaminodiphenyl methane 0.18 g (0.92 mmol), 4,4′-diaminodiphenyl ether 0.18 g (0.92 mmol), and isophthaloyl chloride 0.4 g (1.92 mmol). Off-white solid mass. Yield: 88%; FTIR in υ (cm^−1^): 3271 (m, N-H stretch), 3099, 3045 (w, C-H stretch Ar), 2978, 2942 (w, C-H stretch Al), 1645 (s, amide I), 1599 (s, amide II), 1216 (s, COC), 814 (m, C_6_H_4_ bend); and ^1^H NMR (300 MHz, H_2_SO_4_-d2, δ ppm): 7.81–7.01 (mult. 20H, Ar), 2.8 (sing. 2H, Ar-CH_2_).

I3: 4,4′-Diaminodiphenyl ether 0.18 g (0.92 mmol), ethylene diamine 0.06 g (0.92 mmol), and isophthaloyl chloride 0.4 g (1.92 mmol). Off-white solid mass. Yield: 85%; FTIR in υ (cm^−1^): 3272 (m, N-H stretch), 3114, 3053 (w, C-H stretch Ar), 2970, 2941 (w, C-H stretch Al), 1644 (s, amide I), 1600 (s, amide II), 1216 (s, COC), 826 (m, C_6_H_4_ bend); and ^1^H NMR (300 MHz, H_2_SO_4_-d2, δ ppm): 7.79–6.72 (mult. 16H, Ar), 3.59 (mult. 4H, CH_2_-Al).

Ferrocene-based PDMS-containing block copolymers and their organic analogs: The synthesis of block copolymers is displayed in Figure 3. In a prebaked two-necked flask equipped with a condenser and magnetic stirrer, a mixture of diamines (1.84 mmol) was dissolved in 15 mL of freshly dried THF and treated with 10 mL of triethylamine under an inert atmosphere. The temperature was lowered to 0 °C using an ice bath, and 1.92 mmol of acid chloride (1,1′-ferrocenedicarbonyl chloride/terephthaloyl chloride/isophthaloyl chloride) dissolved in 10 mL of dry THF was added dropwise with vigorous stirring over half an hour. After that, the temperature was slowly raised to room temperature and PDMS (0.02 mmol, 0.1 g) dissolved in 10 mL of dry THF was added dropwise to the reaction mixture. The reaction mixture was then allowed to reflux for 72 h. After completion of the reaction monitored by the TLC, the reaction mixture was cooled to room temperature and poured into an excess of methanol. The precipitated product was then filtered and washed several times with dry methanol, dry THF, and n-hexane. The obtained product was vacuum dried for 24 h.

PF1: 4,4′-Diaminodiphenyl sulfone 0.2 g (0.92 mmol), 4,4′-diaminodiphenyl ether 0.18 g (0.92 mmol), 1,1′-ferrocene dicarbonyl chloride 0.6 g (1.92 mmol), and PDMS (0.02 mmol, 0.1 g). Dark brown solid mass. Yield: 60%; FTIR in υ (cm^−1^): 3306 (m, N-H stretch), 3072, 3033 (w, C-H stretch Ar), 2963, 2939 (w, C-H stretch Al), 1644 (s, amide I), 1593 (s, amide II), 1312, 1140 (s, SO_2_), 1260 (s, Si-C-H), 1214 (s, COC), 1102, 1033 (s, Si-O-Si), 813 (m, C_6_H_4_ bend), 488 (m, Fe-Cp); and ^1^H NMR (300 MHz, DMSO-d6, δ ppm): 7.71–6.98 (mult, 16 H, Ar), 4.97–4.45 (br, 8H, Cp), 2.99 (quart. 2H, NH-CH_2_), 1.76 (pent. 2H, CH_2_), 1.13 (trip. 2H, CH_2_-Si), 0.04 (sing. 6H, Si-CH_3_).

PF2: 4,4′-Diaminodiphenyl methane 0.18 g (0.92 mmol), 4,4′-diaminodiphenyl ether 0.18 g (0.92 mmol), 1,1′-ferrocene dicarbonyl chloride 0.6 g (1.92 mmol), and PDMS (0.02 mmol, 0.1 g). Dark brown solid mass. Yield: 68%; FTIR in υ (cm^−1^): 3279 (m, N-H stretch), 3123, 3064 (w, C-H stretch Ar), 2962, 2866 (w, C-H stretch Al), 1648 (s, amide I), 1613 (s, amide II), 1261 (s, Si-C-H), 1172 (s, COC), 1097, 1021 (s, Si-O-Si), 813 (m, C_6_H_4_ bend), 488 (m, Fe-Cp); and ^1^H NMR (300 MHz, D_2_SO_4_, δ ppm): 5.03–4.45 (br, 8H, Cp), 7.82–6.59 (m, 16 H, Ar), 3.29 (2.50 (s, 2 H, CH_2_), 3.29 (quart. 2H, NH-CH_2_), 2.50 (sing, 2H, CH_2_-Ar), 1.73 (pent. 2H, CH_2_), 0.54 (trip. 2H, CH_2_-Si), 0.04 (sing. 6H, Si-CH_3_).

PF3: 4,4′-Diaminodiphenyl ether 0.18 g (0.92 mmol), ethylene diamine 0.06 g (0.92 mmol), 1,1′- ferrocene dicarbonyl chloride 0.6 g (1.92 mmol), and PDMS (0.02 mmol, 0.1 g). Light brown solid mass. Yield: 65%; FTIR in υ (cm^−1^): 3336 (m, N-H stretch), 3104, 3114 (w, C-H stretch Ar), 2969–2872 (w, C-H stretch Al), 1631 (s, amide I), 1591 (s, amide II), 1261 (s, Si-C-H), 1216 (s, COC), 1100, 1027 (s, Si-O-Si), 824 (m, C_6_H_4_ bend), 484 (m, Fe-Cp); and ^1^H NMR (300 MHz, D_2_SO_4_, δ ppm): 4.95–4.44 (br. 8H, Cp), 7.84–6.64 (mult. 8H, Ar), 3.50 (mult. 4H, CH_2_-Al), 1.18 (trip. 2H, CH_2_), 0.05 (sing. 6H, Si-CH_3_).

PT1: 4,4′-Diaminodiphenyl sulfone 0.2 g (0.92 mmol), 4,4′-diaminodiphenyl ether 0.18 g (0.92 mmol), terephthaloyl chloride 0.4 g (1.92 mmol), and PDMS (0.02 mmol, 0.1 g). Off-white solid mass. Yield: 97%; FTIR in υ (cm^−1^): 3306 m (N-H stretch), 3114, 3043 (w, C-H stretch Ar), 2978, 2945 (w, C-H stretch Al), 1644 (s amide I), 1590 (s, amide II), 1319, 1146 (s, SO_2_), 1251 (s, Si-C-H), 1217 (s, COC), 1104, 1014 (s, Si-O-Si), 828 (m, C_6_H_4_ bend); and ^1^H NMR (300 MHz, DMSO-d6, δ ppm): 7.71–6.98 (mult, 16 H, Ar), 0.04 (sing. Si-CH_3_).

PT2: 4,4′-Diaminodiphenyl methane 0.18 g (0.92 mmol), 4,4′-diaminodiphenyl ether 0.18 g (0.92 mmol), terephthaloyl chloride 0.4 g (1.92 mmol), and PDMS (0.02 mmol, 0.1 g). Off-white solid mass. Yield: 95%; FTIR in υ (cm^−1^): 3294 (m, N-H stretch), 3121, 3041 (w, C-H stretch Ar), 2970, 2945 (w, C-H stretch Al), 1644 (s, amide I), 1597 (s, amide II), 1250 (s, Si-C-H), 1217 (s, COC), 1102, 1015 (s, Si-O-Si), 827 (m, C_6_H_4_ bend); and ^1^H NMR (300 MHz, H_2_SO_4_-d2, δ ppm): 7.76–6.65 (mult. 12H, Ar), 2.25 (sing. 2H, CH_2_-Ar), 0.07 (sing. Si-CH_3_).

PT3: 4,4′-Diaminodiphenyl ether 0.18 g (0.92 mmol), ethylene diamine 0.06 g (0.92 mmol), terephthaloyl chloride 0.4 g (1.92 mmol), and PDMS (0.02 mmol, 0.1 g). Yellow solid mass. Yield: 85%; FTIR in υ (cm^−1^): 3291 (m, N-H stretch), 3121, 3043 (w, C-H stretch Ar), 2979, 2945 (w, C-H stretch Al), 1637 (s, amide I), 1590 (s, amide II), 1260 (s, Si-C-H), 1217 (s, COC), 1101, 1015 (s, Si-O-Si), 826 (m, C_6_H_4_ bend); and ^1^H NMR (300 MHz, H_2_SO_4_-d2, δ ppm): 7.79–6.51 (mult. 8H, Ar), 3.50 (mult. 4H, CH_2_-Al), 0.16 (sing. Si-CH_3_).

PI1: 4,4′-Diaminodiphenyl sulfone 0.23 g (0.92 mmol), 4,4′-diaminodiphenyl ether 0.18 g (0.92 mmol), and isophthaloyl chloride 0.4 g (1.92 mmol). Off-white solid mass. Yield: 89%; FTIR in υ (cm^−1^): 3247 (m, N-H stretch), 3075, 3042 (w, C-H stretch Ar), 2976, 2936 (w, C-H stretch Al), 1651 (s, amide I), 1599 (s, amide II), 1315, 1147 (s, SO_2_), 1260 (s, Si-C-H), 1217 (s, COC), 1104, 1012 (s, Si-O-Si), 826 (m, C_6_H_4_ bend); and ^1^H NMR (300 MHz, H_2_SO_4_-d2, δ ppm): 7.82–7.01 (mult. 16 H, Ar), 0.06 (sing. Si-CH_3_).

PI2: 4,4′-Diaminodiphenyl methane 0.18 g (0.92 mmol), 4,4′-diaminodiphenyl ether 0.18 g (0.92 mmol), and isophthaloyl chloride 0.4 g (1.92 mmol). Off-white solid mass. Yield: 88%; FTIR in υ (cm^−1^): 3270 (m, N-H stretch), 3114, 3039 (w, C-H stretch Ar), 2975, 2939 (w, C-H stretch Al), 1645 (s, amide I), 1599 (s, amide II), 1251 (s, Si-C-H), 1215 (s, COC), 1100, 1013 (s, Si-O-Si), 814 (m, C_6_H_4_ bend); and ^1^H NMR (300 MHz, H_2_SO_4_-d2, δ ppm): 7.86–6.67 (mult. 12H, Ar), 2.25 (sing. 2H, CH_2_-Ar), 0.06 (sing. Si-CH_3_).

PI3: 4,4′-Diaminodiphenyl ether 0.18 g (0.92 mmol), ethylene diamine 0.06 g (0.92 mmol), and isophthaloyl chloride 0.4 g (1.92 mmol). Off-white solid mass. Yield: 85%; FTIR in υ (cm^−1^): 3307 (m, N-H stretch), 3129, 3055 (w, C-H stretch Ar), 2970, 2941 (w, C-H stretch Al), 1649 (s, amide I), 1600 (s, amide II), 1251 (s, Si-C-H), 1217 (s, COC), 1101, 1012 (s, Si-O-Si), 825 (m, C_6_H_4_ bend); ^1^H NMR (300 MHz, H_2_SO_4_-d2, δ ppm): 7.82–6.59 (mult. 8H, Ar), 3.55 (mult. 4H, CH_2_-Al), 0.14 (sing. Si-CH_3_).

Stretch = stretching, bend = bending, s = sharp, m = medium, w = weak, Al = aliphatic, Ar = aromatic, br. = broad, sing. = singlet, trip. = triplet, quart. = quartet, pent. = pentet, mult. = mutliplet.

### 2.2. Physical Characterization

FT-IR spectra of the synthesized material were recorded on a Bio-Rad Excalibur FTIR spectrometer. ^1^H NMR spectra were obtained in DMSO-d_6_ and H_2_SO_4_-d_2_ solvents using BRUKER’s 400 MHz instruments. The weight-average molecular weights of the selected synthesized material were determined using a Brookhaven BI 200S instrument fitted with an argon-ion laser (Coherent Innova) with vertically polarized incident light of wavelength λ = 637 nm and a BI 9000 AT digital correlator. Thermo gravimetric analysis (TGA), differential thermogravimetry (DTG), and differential scanning calorimetry (DSC) were performed on an Sdt Q600 V20.9 Build 20 of Universal V4.5A TA Instruments, at a heating rate of 20 °C per minute under a nitrogen atmosphere. Wide angle X-ray diffraction (WXRD) diffractograms were taken on Phillips 3040/60 X’Pert PRO diffractometer equipped with a Cu-Kα radiation source. The surface morphology and percentage composition of each atom of the samples were determined by scanning electron microscopy (SEM) and energy dispersive X-ray (EDX) techniques, respectively, using the SU-1500 (HITACHI) instrument. The ASTM D570-81 procedure was followed for water absorption studies. The DFT calculations were carried out on Gaussian05/Gaussview09 software by optimization (at DFT/B3LYP/6-311G) followed by a vibrational analysis of monomers and representative polymeric structures by selecting the DFT/B3LYP/6-311G** (d, p) basis set.

## 3. Results and Discussion

### 3.1. Synthesis and Characterization

1,1′-ferrocene dicarbonyl chloride: The ferrocene monomer, 1,1′-ferrocene dicarbonyl chloride, was prepared in three steps as shown in Figure 1. Briefly, in the first step 1,1′ diacetyl ferrocene was prepared using acetyl chloride in the presence of a catalyst AlCl_3_ as reported earlier [29]. In the second step, the acidification of the 1,1′-diacetyl ferrocene was carried out using sodium hypochlorite, which resulted in the formation of 1,1′ ferrocene dicarboxylic acid [30]. In the third step, 1,1′ ferrocene dicarboxylic acid was converted into the more reactive monomer, 1,1′-ferrocene dicarbonyl chloride, using a more efficient method as previously reported by our group [18].

Ferrocene-based terpolyamides and their organic analogs: Ferrocene-based aromatic terpolyamides were synthesized using different combinations of aromatic diamines with 1,1′-ferrocene dicarbonyl chloride via a low-temperature solution polycondensation method under inert conditions (nitrogen atmosphere) as shown in Figure 2. The synthesis is followed by nucleophilic substitution in which the nucleophilic amine (NH) attacks the electrophilic carbon of carbonyl (C=O) moiety having a good leaving group (Cl^−^). The monomer ratio used was 1:0.5:0.5 (1,1′-ferrocene dicarbonyl chloride: diamine 1: diamine 2). All the ferrocene-based terpolyamides were solid brown powders with a high yield (up to 95%). The various combinations of diamines including ether, sulfone, and methylene linkages were used to draw a structure–property relationship. The organic analogs of the terpolyamides were prepared for comparative studies using terephthaloyl chloride and isophthaloyl chloride by the same procedure. All the organic analogs were off-white solids with high yields like their corresponding ferrocene-based terpolyamides. All the synthesized terpolyamides were structurally elucidated by FTIR and ^1^H NMR spectroscopic techniques.

The structures of the synthesized ferrocene-based terpolyamides and their organic analogs were elucidated using FTIR and ^1^H NMR spectroscopic techniques. Appendix A shows the representative FTIR spectra of the ferrocene-based terpolyamide and its block copolymer, respectively. The FTIR spectra of the organic analogs are displayed in Appendix A. The FTIR spectra depict the occurrence of the polycondensation for the synthesis of terpolyamides by the presence of strong, characteristic stretching vibration bands for the amide group in the ranges of 3350–3270 cm^−1^ (N-H stretch), 1668–1633 cm^−1^ (amide carbonyl), and 1600–1540 cm^−1^ (N-H deformation) that is according with the reported literature [17]. The stretching bands of weak intensities were observed in the range of 3000–2900 cm^−1^ for the aliphatic sp^3^-CH moieties. The aromatic sp^2^-CH appeared in the 3000–3100 cm^−1^ range. The presence of ether linkage was shown by the strong stretching band in the region of 1270–1235 cm^−1^. Two characteristic peaks appeared for sulfone linkage in the ranges of 1325–1310 cm^−1^ and 1150–1145 cm^−1^. A distinguishing stretching band of medium intensity for the cyclopentadienyl ring of the ferrocene (Fe-Cp) was shown at 490–480 cm^−1^ which was found absent in the FTIR spectra of the corresponding organic analogs.

The ^1^H NMR spectra of the ferrocene-based terpolyamides were recorded using deuterated DMSO (DMSO-d6) while deuterated sulfuric acid (H_2_SO_4_-d2) was used for their organic analogues. All the characteristic peaks for aromatic and aliphatic protons were observed in the ranges of 7.84–6.34 ppm and 1.88–2.20 ppm, respectively. A distinguishing broad peak of Fe-Cp protons from their organic analogs appeared in the region of 5.02–4.49 ppm. A representative ^1^H NMR spectrum of the ferrocene-based terpolyamide F2 is shown in Appendix A. The representative ^1^H NMR spectra for organic analogs are displayed in Appendix A.

Ferrocene-based PDMS-containing block copolymers and their organic analogs: Aminopropyl-terminated polydimethylsiloxane (PDMS) was used to synthesize ferrocene-based block copolymers and their corresponding organic analogs (Figure 3). A one-pot two-step solution polycondensation method was used for the synthesis of block copolymers. A molar ratio of 1:0.96:0.4 was maintained for 1,1′-ferrocene dicarbonyl chloride, the combination of diamines (0.96 = 0.48 + 0.48), and PDMS, respectively. The temperature of the reaction was initially maintained at 0 °C to avoid side reactions of the highly reactive acyl chlorides. Triethylamine was used as an acid scavenger to speed up the reaction by abstracting the proton to form an escapable HCl byproduct [31].

The FTIR spectra of the corresponding block copolymers exhibited all the characteristic bands found in terpolyamides. The appearance of characteristic stretching vibrations in the regions of 1260 cm^−1^ (Si-CH_3_) and 1105–1015 cm^−1^ (Si-O-Si doublet maxima) confirmed the presence of a PDMS block in all the copolymers [32]. A stretching band for C-O moiety present in the PDMS block was observed as a small shoulder in the region of 1217 cm^−1^. An example of the FTIR spectrum of the ferrocene-based PDMS-containing block copolymer of the sample is shown in Appendix A while the representative FTIR spectra of the organic analogs are given in Appendix A.

The ^1^HNMR spectra of the block copolymers exhibited all the characteristic peaks in the corresponding regions found for terpolyamides with some additional peaks for siloxane moiety. The signal corresponding to the protons Si-CH_3_ appeared as a singlet at 0.04 ppm while the rest of the methylene protons’ (CH_2_) peaks were observed in the region of 1.16–2.96 ppm. The terminal NH proton appeared at a chemical shift of 3.5 to 4.0 ppm. Aromatic protons showed characteristic chemical shifts in the range of 6.3 to 7.8 ppm while Fe-Cp ring protons displayed a chemical shift value in the region of 5.03–4.45 ppm. Appendix A shows the representative ^1^H NMR spectrum of PF2. The representative ^1^H NMR spectra of the organic analogs of PDMS-containing block copolymers are shown in Appendix A.

### 3.2. Solubility

The qualitative solubility data of the synthesized terpolyamides, their block copolymers, and their corresponding organic analogs are given in Table 1. Ferrocene-based terpolyamides and their block copolymers showed improved solubility in common polar aprotic solvents such as dimethyl sulfoxide (DMSO), dimethylformamide (DMF), and N-methyl-2-pyrrolidone (NMP). This improved solubility of these materials might be due to the incorporation of flexible units such as -SO_2_-, -O-, and -CH_2_-, and kinked linkages such as Fe-Cp and NHCO. However, their organic analogs were insoluble in common organic solvents, which can be attributed to their chain stiffness, rigid amide group, and strong hydrogen bonding [32]. The incorporation of siloxane moiety has no pronounced effect on the solubility behavior. This may be due to the increased molecular weight of the block copolymer as reported earlier [32]. All the synthesized materials were soluble in concentrated H_2_SO_4_ due to their higher degree of hydrogen bonding affinity with H_2_SO_4_ than among the polymer chains. A recent study has also suggested that H_2_SO_4_ can be used as an alternative solvent to water [33].

### 3.3. Molecular Weight Determination

The static laser light scattering (sLLS) technique was used to investigate the molecular weights of the selected soluble terpolyamide F2 and its block copolymer PF2, in DMSO at room temperature. The sLLS technique measures the light intensity as a function of solute concentration and scattering angle. Zimm’s formalism can be used to describe a relationship between polymer solution concentration and weight average molecular weight through the determination of the radius of gyration, average molecular weight, and second virial coefficient. F2 showed a high molecular weight of 7.5 × 10^6^ g/mol indicating that the solution polycondensation method is suitable for ferrocene-based polymers as reported previously [32]. The radius of gyration <Rg> reflects the size of the polymers and its value increases with an increase in molecular weight, while the second virial co-efficient (A2) value indicates a strong polymer–solvent interaction [34]. The values of <Rg> and A2 for F2 were found to be 170 nm and 5.86 × 10^−4^ cm^3^ molg^−2^, respectively. The molecular weight of the respective block copolymer PF2 (8 × 10^6^ g/mol) was high as compared to F2 and is comparable to the reported PDMS-containing block copolymers [35,36,37]. These high values can be attributed to the incorporation of a high molecular weight PDMS segment in the block copolymer. There is an increase in the value of <Rg> (180 nm) and A2 (2.9 × 10^−5^ cm^3^ molg^−2^) displaying longer chain lengths and a strong polymer–solvent interaction.

### 3.4. Wide Angle X-ray Diffraction

The morphology of the synthesized polymeric material was examined using a wide-angle X-ray diffraction (WXRD) pattern in the region of 2θ = 0°–70° at room temperature. The representative diffractograms of the polymers are shown in Figure 4. The broad peaks were observed for all the polymers which illustrated the amorphous nature of the synthesized material. The diffused peaks for ferrocene-based terpolyamides depicted the incorporation of the ferrocene moiety in the polymer chains which interrupted the close packing of the polymer chains leading to their enhanced solubility and decreased crystallinity. Among organic analogs, terephthaloyl-based terpolyamides showed more crystalline nature as compared to isophthaloyl-based terpolyamides. This can be attributed to the para–para symmetry in the polymer chains leading toward their close packing, while the introduction of the meta orientation in isophthaloyl-based terpolyamides distorts the polymer lattice [38]. The broad peak patterns were observed for the PDMS-containing block copolymers which can be due to the introduction of siloxane moiety in the polymer backbone that reduces the crystallinity [39,40].

### 3.5. Surface Morphology

Scanning electron microscopy (SEM) is considered the most widely used technique in modern laboratories for surface analysis and, therefore, it was utilized to analyze the surface morphology of the synthesized polymeric materials. The surface morphology of the ferrocene-based aromatic terpolyamides showed a sponge-like appearance along with cavities which indicate the modification of the polymer surface by the metal atom (Figure 5). The PDMS-containing block copolymers showed diffused morphology that can be attributed to the movement of siloxane moiety toward the surface of the material as reported earlier [31,38]. The siloxane moiety can become a dominant part of the surface and, due to this unique feature of siloxane, these materials can also be used as surface modifiers [17,18,41].

The EDX patterns of the synthesized terpolymers and their block copolymers were also taken along with their SEM images on the same instrument to provide evidence for the presence of iron and silicon atoms. Figure 6 shows that the EDX patterns of the representative terpolymer and block copolymer confirmed the presence of the iron and silicon atoms in their structure, respectivel

### 3.6. Thermal Properties

The thermal stability of the synthesized polymeric material was studied using thermogravimetric analysis (TGA), differential thermogravimetry (DTG), and differential scanning calorimetry (DSC). For TGA studies, the synthesized polymeric material was heated from room temperature to 800 °C at a heating rate of 10 °C/min under a nitrogen atmosphere. The glass transition temperature (T_g_), the decomposition temperature at 10% weight loss (T_10_), the temperature at maximum weight loss (T_max_), the final decomposition temperature at the end of the experiment (T_f_), the percentage char yield (CR), and limiting oxygen index (LOI) values are summarized in Table 2.

Figure 7A,B show the TGA, DSC, and DTA data of the representative terpolyamide F3 and its block copolymer PF3, respectively. Figure 7A,B show an exothermic peak with 20% weight loss at 200 °C that can be assigned to the bound moisture or solvents with the terpolyamides and block copolymers, respectively. The TGA data reveal that all the ferrocene-based polymers showed no decomposition below 220 °C, while their corresponding analogs were found to be stable up to 190 °C. The temperature for 50% weight loss was seen in the range of 450 °C to 680 °C for terpolyamides while their block copolymers showed maximum degradation in a range from 400 °C to 600 °C. In the case of block copolymers, multiple exothermic peaks were observed up to 600 °C which can be due to the decomposition of siloxane moiety to silica-based chars. It can be seen from the data that ferrocene-based terpolyamides have high thermal stability as compared to their organic analogs. This can be attributed to the presence of metal atoms in the backbone of the polymeric chain [42]. The terephthaloyl-based terpolyamides were found to be more thermally stable as compared to the isophthaloyl-based terpolyamides. It may be due to the more ordered nature of terephthaloyl moiety that leads to the close packing of chains. However, in the case of isophthaloyl moiety, the symmetry is disturbed due to meta catenation which leads to the less packing of polymeric chains; hence, they are less thermally stable [18].

The percentage char yield of ferrocene-based terpolyamides was found to be higher than their organic analogs due to the presence of metal atoms in its structure. The terpolyamides containing sulfone linkages were found to be more thermally stable than their methylene-containing analogs which matches the literature findings [43]. When comparing the thermal stability of methylene-containing terpolyamides attached to the aromatic ring with aliphatic methylene groups, it was observed that aromatic methylene showed higher thermal stability than aliphatic methylene. This can be attributed to the fact that the aromatic ring needs more thermal energy to decompose as compared to the aliphatic C-H bond [17]. 

The terpolyamides and their block copolymers showed high glass transition temperatures that were comparable to the reported literature [32]. All the synthesized polymeric materials exhibited high thermal stability which can be attributed to the intermolecular hydrogen bonding responsible for chain stiffness.

The thermal stability of the block copolymers was found to be comparable to the synthesized terpolyamides which is a reverse trend, as siloxane moiety lowers the thermal stability due to its soft nature [18]. This may be due to the higher molecular weight of the block copolymers as compared to their corresponding terpolyamides. A high percentage char yield of block copolymers was observed which may be because siloxane moiety makes some ceramic-like precursors, yielding high char yield [35].

The flammability of a polymer generally depicts its thermal stability [44]. The flame retardation behavior was studied using thermal degradation data to calculate the limiting oxygen index (LOI). The Van Krevelen–Hoftyzer equation was used to calculate the LOI values of the synthesized materials using their char yield data obtained from their TGA thermograms. The LOI data of the synthesized materials fall in the range of 24.3–33.5, which shows their good flame retardation behavior according to the reported literature [44,45]. Ferrocene-based terpolyamide F1 shows the highest flame retardation behavior (high LOI value) which can be attributed to the presence of sulfur atoms in the sulfone linkage. Sulfur is considered a flame-retardant moiety that releases incombustible SO_2_ and improves char yield as reported previously [46].

### 3.7. Water Absorption Studies

The ASTM D570-81 procedure was followed to study the water absorption behavior of the selected terpolyamides and their block copolymers. The data obtained for water absorption studies are summarized in Table 3. The PDMS-containing block copolymers showed very little water absorption in the range of 0.1–0.5% as compared to their corresponding terpolyamides (25–50%). The presence of polar amide linkages on the surface of the terpolyamides can be responsible for the increased water absorption due to their increased hydrogen bonding with water molecules leading to more hydrophilicity [32]. The low water absorption by the block copolymers can be attributed to the great hydrophobic character of siloxane moiety that obscures the polar amide linkages on the surface of the block copolymer [41].

### 3.8. DFT Calculations

DFT calculations were performed to elucidate the role of monomers in the synthesis of terpolymer and block copolymers. The results obtained from DFT calculations are summarized in Table 4 and Table 5, and Figure 8. The monomers’ information regarding their codes, structures, HOMO, and LUMO orbital structures is displayed in Appendix A. The computed electronic parameters, dipole moments, and imaginary vibrational frequencies are shown in Table 4. It is clear from the data displayed in Table 4 that the HOMO energies predict stabilizing nature for the outermost electrons of M1, M1′, M2, M3, M4, and M5 while the HOMO is less stale for monomers M6 and M7. This reveals the transfer of an electron to be readily feasible from HOMO of M6 and M7. On the other hand, the LUMO orbital of M6 and M7 are not stabilizing for any incoming electrons. The negative values of HOMO orbitals for M1, M1′, M2, M3, M4, and M5 reveal their stabilizing nature. The total energies of polymers F2 (M5 + M6), PF2 (M5 + M6 + M7), T2 (M1′ + M5), and PT2 (M1′ + M6 +M7) are more negative in value than their respective monomers mentioned in the bracket. This shows that the stability of the HOMO orbitals is achieved with the polymerization of F2, PF2, T2, and PT2. The computed dipole moments reveal M3 to be the most polar among the other monomers.

The imaginary frequencies were obtained during the vibrational computation of M1, M2, and M4 which can be used for the predictions of the functional group involved in the formation of the transition state during the polymerization process. In M1, the Cl (chloride) bending vibrations are associated with imaginary frequencies which predict the involvement of COCl in the transition state formation in electron pair acceptance behavior. The appearance of two imaginary frequencies for the H symmetric bending mode for end groups in M2 and three NH angle bending modes in M4 reveal the transition state involving terminal groups in M2 and the whole molecule in M4, which is complemented by the LUMO spread in M4 as depicted in Table 4. The stabilizing LUMO which can accept the incoming electron pair has iso-densities spread over the whole molecule justifying the contribution of the molecule as a whole towards transition state formation in any electron transfer reaction.

The polymerization in the cases of T2 (M1′ + M5) and F2 (M5 + M6) are presented in terms of the electron pair donor–acceptor relation in Figure 8. The DFT calculation based on the relative energies for the constituents of T2 and F2 predicts the electron pair donation from HOMO of M5 to LUMO of M1′ and M6, leading to the formation of T2 and F2, respectively. However, the energy gap between the LUMO of M5 to HOMO of both M1′ and M6 prohibited the transfer of an electron from M1′ or M6 towards M5.

Table 5 displays the thermodynamic parameters of the monomers where M6 is found to be associated with maximum enthalpy due to its large size while the smallest monomer M2 depicts the lowest enthalpy. The zero-point vibrational energies reveal M1′ to be the least hindered and M7 to be the most, because of its bulky nature. The heat capacities and entropy are also given in Table 5.

## 4. Conclusions

The structural features of all the synthesized materials were confirmed by using the FTIR and ^1^H NMR spectroscopic techniques. Ferrocene-based terpolyamides and their corresponding block copolymers were found to be soluble in common polar aprotic solvents, such as DMSO and DMF, while their organic analogs were insoluble in all the tested solvents. All the synthesized materials were soluble in H_2_SO_4_. The terpolyamides and block copolymers showed high thermal stability assessed by their TGA, DTG, and DSC analyses. The thermal degradation of ferrocene-based materials below 300 °C was not observed. A high percent char yield was obtained for ferrocene-based terpolyamides and their corresponding block copolymers, which was further used to measure the limiting oxygen index (LOI), illustrating their good flame-retardant behavior. The SEM analysis exhibited the sponge-like and diffused morphology of the ferrocene-based terpolyamides and their block copolymers, respectively. EDX spectra depicted the presence of Fe and Si atoms in the respective materials. The water absorption study of the selected terpolyamides and their block copolymers has revealed that ferrocene-based terpolyamide absorbs higher water content as compared to its PDMS-containing block copolymer. The DFT calculation predicted the electron pair donation from HOMO of M5 to LUMO of M′ and M6 leading to the formation of T2 and F2, respectively. Further, the involvement of the terminal functional groups of M1 and M2 was predicted from computed imaginary frequencies, while M4 displayed the whole molecule involvement in the polymerization process. Overall, the introduction of organometallic ferrocene moiety along with variable flexible units such as -SO_2_-, -O-, and -CH_2_- improved the solubility as well as the processability of the rigid polyamides without compromising their exceptional properties. Hence, the approach is useful in removing the barrier in the applicability of such materials.

## Figures and Tables

**Figure 1 polymers-14-05087-f001:**
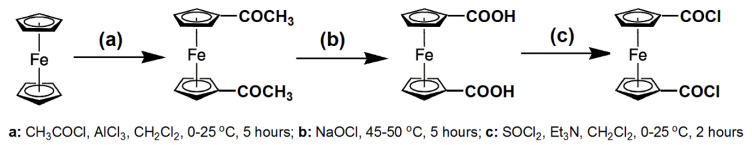
Synthesis of 1,1′-ferrocenedicarbonyl chloride.

**Figure 2 polymers-14-05087-f002:**
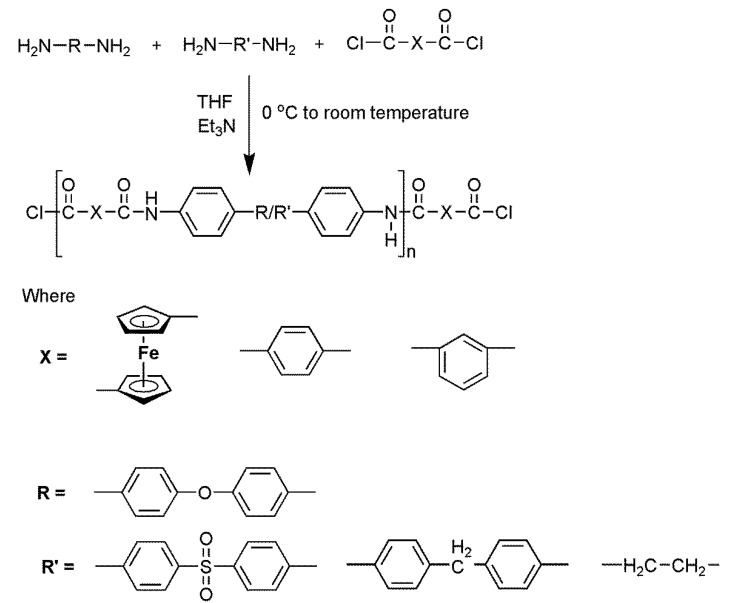
Synthesis of ferrocene-based aromatic terpolyamides and their organic analogs.

**Figure 3 polymers-14-05087-f003:**
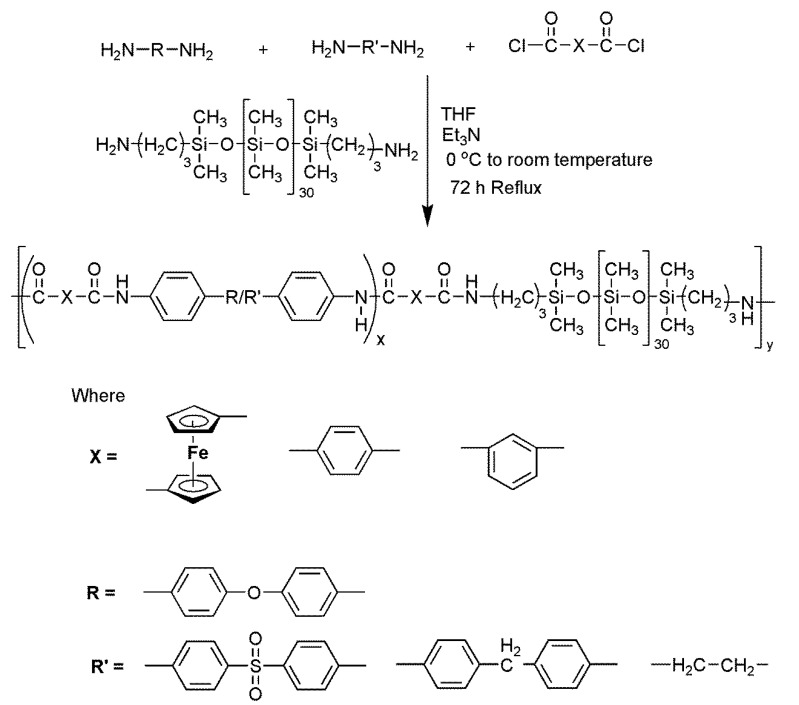
Synthesis of ferrocene-based PDMS-containing block copolymers and their organic analogs.

**Figure 4 polymers-14-05087-f004:**
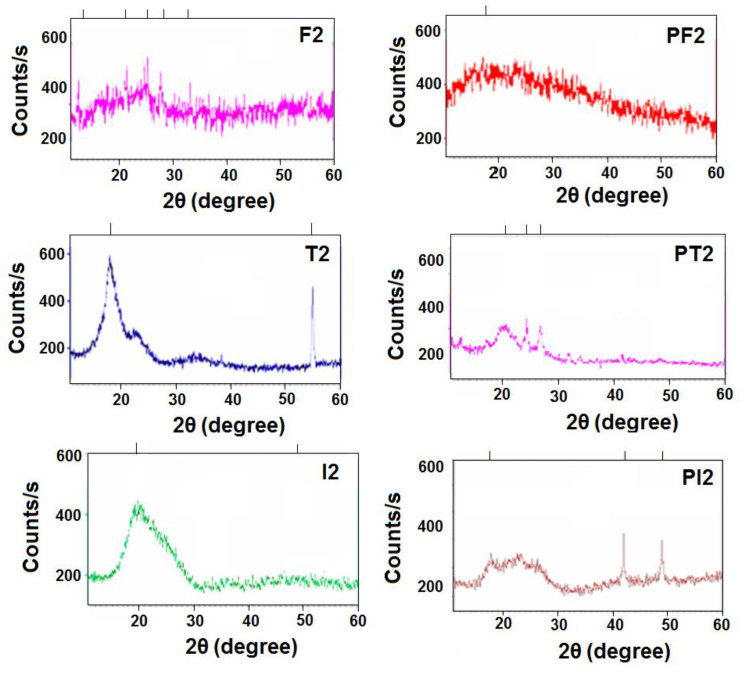
WXRD patterns of the representative terpolyamides and their corresponding block copolymers.

**Figure 5 polymers-14-05087-f005:**
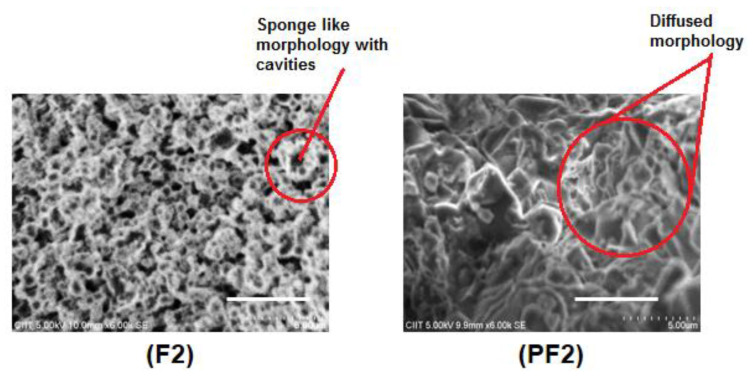
SEM images of the representative terpolyamides and their block copolymers, scale bar: 5 μm.

**Figure 6 polymers-14-05087-f006:**
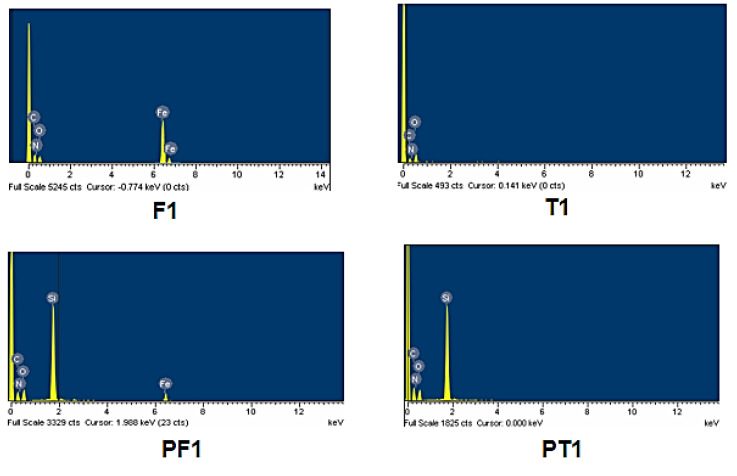
EDX spectra of representative terpolyamides and block copolymers.

**Figure 7 polymers-14-05087-f007:**
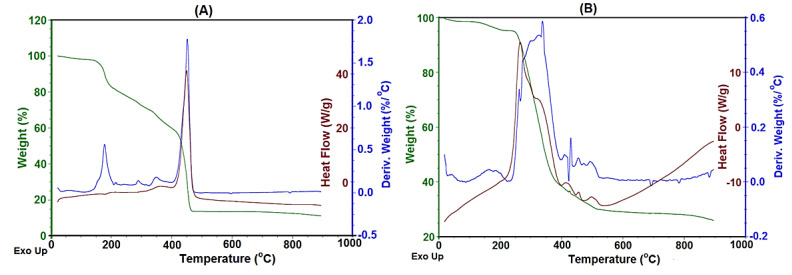
Thermograms of representative terpolyamide F3 (**A**) and block copolymer PF3 (**B**). TGA green lines, DTG (blue lines), and DSC (red lines).

**Figure 8 polymers-14-05087-f008:**
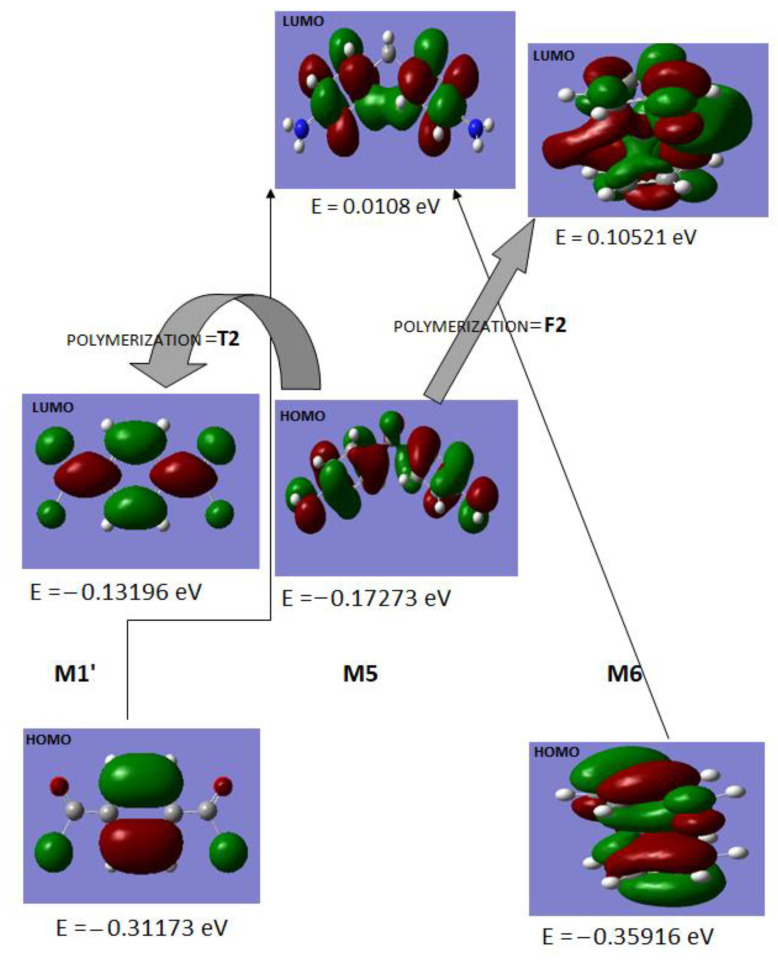
Computed HOMO and LUMO spread of M1′, M5, and M6 combining to form T2 (M1′ + M5) and F2 (M5 + M6) at DFT/B3LYP/6-311G** (d, p) basis set.

**Table 1 polymers-14-05087-t001:** Solubility data of the ferrocene-based terpolyamides, PDMS-containing block copolymers, and their organic analogs.

Solvents	F1	F2	F3	T1	T2	T3	I1	I2	I3	PF1	PF2	PF3	PT1	PT2	PT3	PI1	PI2	PI3
DMSO	++	++	++	--	--	++	--	++	++	++	+-	+-	+-	--	--	+-	+-	+-
DMF	+-	+-	+-	--	--	--	--	--	--	+-	--	+--	--	--	--	--	--	--
THF	+-	++	+-	--	--	--	--	--	--	+-	--	+-	--	--	--	--	--	--
m-Cresol	--	+-	+-	--	--	--	--	--	--	+-	--	+-	--	--	--	--	--	--
NMP	++	++	++	--	--	--	--	++	+-	++	--	++	--	--	--	--	++	+-
CHCl_3_	--	--	--	--	--	--	--	--	--	--	--	--	--	--	--	--	--	--
CH_2_Cl_2_	--	--	--	--	--	--	--	--	--	--	--	--	--	--	--	--	--	--
Acetone	--	--	--	--	--	--	--	--	--	--	--	--	--	--	--	--	--	--
CH_3_CN	--	--	--	--	--	--	--	--	--	--	--	--	--	--	--	--	--	--
CH_3_OH	--	--	--	--	--	--	--	--	--	--	--	--	--	--	--	--	--	--
C_2_H_5_OH	++	--	++	--	--	--	--	--	--	++	--	--	--	--	--	--	--	--
(C_2_H_5_)_2_O	--	--	--	--	--	--	--	--	--	--	--	--	--	--	--	--	--	--
EtOAc	--	--	--	--	--	--	--	--	--	--	--	--	--	--	--	--	--	--
n-C_6_H_14_	--	--	--	--	--	--	--	--	--	--	--	--	--	--	--	--	--	--
Toluene	--	--	--	--	--	--	--	--	--	--	--	--	--	--	--	--	--	--
H_2_SO_4_	++	++	++	++	++	++	++	++	++	++	++	++	++	++	++	++	++	++

-- completely insoluble, +- soluble on heating, ++ soluble at room temperature, DMSO = dimethylsulphoxide, DMF = N,N′-dimethylformamide, THF = tetrahydrofuran, NMP = N-methyl-2-pyrrolidone, CHCl_3_ = chloroform, CH_2_Cl_2_ = dichloromethane, CH_3_CN = acetonitrile, CH_3_OH = methanol, C_2_H_5_OH = ethanol, (C_2_H_5_)_2_O = diethylether, EtOAc = ethyl acetate, n-C_6_H_14_ = n-hexane, and H_2_SO_4_ = sulphuric acid.

**Table 2 polymers-14-05087-t002:** Thermal properties of the ferrocene-based terpolyamides, block copolymers, and their corresponding organic analogs.

Sample Codes	Tg (°C)	T_10_ (°C)	T_max_ (°C)	T_f_ (°C)	CR	LOI
F1	270	280	580	830	31	29.9
F2	220	230	510	710	27	28.3
F3	210	220	440	765	15	23.5
T1	210	220	580	780	31	29.9
T2	190	210	520	801	29	29.1
T3	185	193	435	685	10	21.5
I1	186	192	560	804	01	17.9
I2	185	200	5100	848	02	18.3
I3	175	190	500	854	01	17.9
PF1	220	240	600	834	40	33.5
PF2	200	280	440	581	30	29.5
PF3	190	210	400	550	25	27.5
PT1	200	230	530	826	38	32.7
PT2	180	200	550	762	35	31.5
PT3	185	192	480	811	28	28.7
PI1	190	198	510	846	30	29.5
PI2	170	195	640	797	29	29.1
PI3	169	190	450	800	23	26.7

T_10_ = Temperature at 10% weight loss, T_max_ = temperature at maximum weight loss, T_f_ = final temperature at the end of curve, CR = % char yield, limiting oxygen index (LOI) = 17.5 + 0.4 (CR) [46].

**Table 3 polymers-14-05087-t003:** Water absorption data of the selected terpolyamides and their block copolymers.

Terpolyamides	% W_A_ ^a^	Block Copolymers	% W_A_ ^a^
F2	25	PF2	0.1
T2	50	PT2	0.5

^a^ %W_A_ = [(W_f_ − Wo)/W_o_] × 100.

**Table 4 polymers-14-05087-t004:** Computed electronic parameters, polarity, and vibrational energies of monomers and representative polymers at DFT/B3LYP/6-311G** (d, p).

Monomers	E_HOMO_ (eV)	E_LUMO_ (eV)	Total Energy (a.u.)	Dipole Moment (Debye)	Imaginary Frequency (cm^−1^)
M1	−0.38812	−0.03531	−1367.012	5.3808	−21.06 -Cl Bending vibrations
M1′	−0.31173	−0.13196	−1371.183	1.0527	--
M2	−0.33884	−0.33884	−188.213	0.0086	−215.20 (-H symmetric bending mode 1) −200.05 (-H symmetric bending mode 2)
M3	−0.20167	−0.20167	−1116.536	8.8394	Nil
M4	−0.20966	−0.00810	−645.619	1.5962	−624.73 (-HN1 angle bending) −545.52 (-HN2 angle bending) −50.43 (M4 symmetric bending)
M5	−0.17273	0.01828	−609.956	2.2372	--
M6	0.10521	0.10521	−1862.783	1.2979	--
M7	−0.31548	0.15462	−1429.508	5.355	--
F2	−0.3221	0.31617	−4513.094	1.287	--
T2	−0.2152	0.22139	−4892.113	1.114	--
PF2	−0.1872	0.35460	−5732.652	4.232	--
PT2	−0.2238	0.11828	−6212.342	4.338	--

**Table 5 polymers-14-05087-t005:** Computed thermodynamic parameters of monomers and representative polymers at DFT/B3LYP/6-311G** (d, p) basis set.

Monomers	Enthalpy (kcal/mol)	Heat Capacity (Ca/mol-K)	Entropy (Ca/mol/K)	Zero-Point Vibrational Energy (kcal/mol)	Total Thermal Energy (kcal/mol)
M1	−1366.896	31.845	93.541	66.735	72.017
M1′	−1371.070	37.492	104.295	63.833	70.379
M2	−188.090	15.274	68.483	72.736	75.634
M3	−1116.295	60.650	126.775	140.955	150.697
M4	−645.388	45.903	106.120	136.397	147.474
M5	−609.696	53.31	120.813	152.976	161.469
M6	−1862.456	49.90	111.356	119.420	127.531
M7	−2165.322	48.71	152.244	172.34	132.560

## Data Availability

The data that support the reported results are presented in the manuscript and Supporting Information using text, figures, and tables.

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
