# Peer review of "Ferrocene-Based Terpolyamides and Their PDMS-Containing Block Copolymers: Synthesis and Physical Properties"

_polymers, 2022, doi:10.3390/polym14235087_

Round 1

Reviewer 1 Report

The manuscript is of scientific interest, but this should be major revised or rejected.

It is unclear, why did authors selected nitrogen protective atmosphere for their studies.

It is unclear, why did authors in Materials and Methods describe not all the applied methods of studying, but then add the required descriptions in other paragraphs? This should be corrected.

There are some misrpints. Authors should carefully check all the text.

3.4. Wide-angle X-ray diffraction. Authors write: "at room temperature (Figure 4). The representative
diffractograms of the polymers are shown in Figure 6.
" This should be corrected. Numbers of figures should be sequential. Fig. 4 should be followed by fig. 5.

Also, the same place. Fig. 6 does not show diffractograms. There are EDS-spectra. This should be corrected and all similar places should be carefully checked.

Authors provide WXRD-patterns in fig. 4. There is no area of small-angle scattering, which is almost always observed in XRD-patterns collected from 2theta 0-7 deg. Authors should explain this.

Fig. 4. There are some problem with axis: x axis represents two ranges 0-35 and 35-70 deg. But sections on the X axis are not equal. It is impossible. Y axis does not provide information about relative intensity. This should be corrected.

Phrase about amorphous peaks for all the phases is not correct, as XRD-patterns show few narrow peaks.

Phrase "incorporation of the ferro-
cene moiety in the polymer chains that interrupted the close packing of the polymer chains
" is not proven. This is only assumption, which should be checked and proved by other methods in addition to XRD.

Authors write "The surface morphol-
ogy of the ferrocene-based aromatic terpolyamides showed a sponge-like appearance
along with cavities which indicates the modification of the polymer surface by the metal
atom
" but but there are no visible cavities modyfied by metal. At the same time, EDX spectra do not prove anything, as fig. 5 does not show areas, where EDX was measured. Authors should improve their illustraive material.

Diffused morphology is also can not be seen in fig. 5 without additional arrows or markers.

Fig. 8 in text appears after fig. 9. This should be corrected.

Author Response

Reviewer’s comment: It is unclear, why did authors selected nitrogen protective atmosphere for their studies.

Authors response: The nitrogen atmosphere is used to avoid any side reactions and the literature reference has been given that also used inert conditions (page 12, line 4).

Reviewer’s comment: It is unclear, why did authors in Materials and Methods describe not all the applied methods of studying, but then add the required descriptions in other paragraphs? This should be corrected.

Authors response: We thank this reviewer for his/her keen observation. One of the missing method was static laser light scattering. It is now added in the physical methods sections on page 11 line # 3.

 Reviewer’s comment: There are some misrpints. Authors should carefully check all the text.

Authors response: We have corrected some misprints. For example, there was one misprint in the surface Morphology section.

Reviewer’s comment: Wide-angle X-ray diffraction. Authors write: "at room temperature (Figure 4). The representative diffractograms of the polymers are shown in Figure 6." This should be corrected. Numbers of figures should be sequential. Fig. 4 should be followed by fig. 5. Also, the same place. Fig. 6 does not show diffractograms. There are EDS-spectra. This should be corrected and all similar places should be carefully checked.

Authors response: Thank you for pointing out this. The figure numbers are corrected and double checked again. These are now in the right sequence.

Reviewer’s comment: Authors provide WXRD-patterns in fig. 4. There is no area of small-angle scattering, which is almost always observed in XRD-patterns collected from 2theta 0-7 deg. Authors should explain this. Fig. 4. There are some problem with axis: x axis represents two ranges 0-35 and 35-70 deg. But sections on the X axis are not equal. It is impossible. Y axis does not provide information about relative intensity. This should be corrected. Phrase about amorphous peaks for all the phases is not correct, as XRD-patterns show few narrow peaks.

Authors response: New data are now added to Figure 4 for XRD patterns to correlate with the text.

Reviewer’s comment: Phrase "incorporation of the ferrocene moiety in the polymer chains that interrupted the close packing of the polymer chains" is not proven. This is only assumption, which should be checked and proved by other methods in addition to XRD.

Authors response: A reference is provided from the literature that also shows this behaviour.

Reviewer’s comment: Authors write "The surface morphology of the ferrocene-based aromatic terpolyamides showed a sponge-like appearance
along with cavities which indicates the modification of the polymer surface by the metal
atom" but but there are no visible cavities modyfied by metal. At the same time, EDX spectra do not prove anything, as fig. 5 does not show areas, where EDX was measured. Authors should improve their illustraive material. Diffused morphology is also cannot be seen in fig. 5 without additional arrows or markers.

Authors response: The SEM images has been improved and new Figure 5 been displayed in the manuscript showing the markers for their sponge-like and diffused morphologies. EDX spectra were just taken to confirm the presence of iron and silicon atoms in the sample. It was run during the SEM analysis of the sample. IT showed the presence of peaks for iron and silicon atoms that are displayed in Figure 6. Also the reference has been given that shows the same studies.

Reviewer’s comment: Fig. 8 in text appears after fig. 9. This should be corrected.

Authors response: It is now corrected. There is no Figure 9.

Reviewer 2 Report

The author of paper which tile is “Ferrocene-Based Terpolyamides and Their PDMS Containing Block Copolymers: Synthesis and Physical Properties” evaluate the on the synthesis of processable ferrocene based terpolyamides and their polydimethylsiloxane (PDMS) containing block copolymers and evaluated its solubility. The Manuscript has its merits, but it required some further enhancements before it can be accepted for publication.

1- The abstract of the paper, please bring more quantitative results in terms of solubility of the synthesized polymer.

2-Introduction of the paper is well-written, however, authors need to bring the previous data in the field and highlight more the novelty of present work. In terms of the application and advantage of the current study, authors should bring more information.

3-Increase the quality of Figure 1. It seems it comes from other sources, if yes, cite the paper in the caption or draw the chemical bonding structure again.

4-In terms of FTIR analysis, where is Fig. S1 or S2 (also NMR results)? I do not have access to the supporting information. It seems its missing. Moreover, for the chemical bonding position of the samples, please cite the reference paper or book.

5-In terms of solubility table, following the data is very hard due to the high number of samples. Please re-organize the data to make them much easier to understand. For example, authors can provide a table for F, T, I, PF, PT, and PI samples and put the concentration of each sample in a single table.  

6-Quality of Fig. 4 and 5 is very low. Please also label all the X-ray peaks for different samples.

7-The EDS results in Figure 6 is not giving us any information at the present format. Authors need to bring the quantitative results in a separate table and explain the results in more-details.

Minor problems:

-        The is a type in the X-ray diffraction part line 3, it should be Figure 4 instead of 4.

Author Response

Comments and Suggestions for Authors

The author of paper which tile is “Ferrocene-Based Terpolyamides and Their PDMS Containing Block Copolymers: Synthesis and Physical Properties” evaluate the on the synthesis of processable ferrocene based terpolyamides and their polydimethylsiloxane (PDMS) containing block copolymers and evaluated its solubility. The Manuscript has its merits, but it required some further enhancements before it can be accepted for publication.

Reviewer’s comment: The abstract of the paper, please bring more quantitative results in terms of solubility of the synthesized polymer.

Authors response: The solubility is briefly mentioned in the abstract. However, the abstract is already quite long and it will not be appropriate to add more text.

Reviewer’s comment: Introduction of the paper is well-written, however, authors need to bring the previous data in the field and highlight more the novelty of present work. In terms of the application and advantage of the current study, authors should bring more information.

Authors response: Thanks for your kind words. We have added more recent references in the introduction to make the work more influential.

Reviewer’s comment: Increase the quality of Figure 1. It seems it comes from other sources, if yes, cite the paper in the caption or draw the chemical bonding structure again.

Authors response: Figure 1 has been replaced with new one that is made again in ChemDraw ultra 8.0 software and saved the image with high resolution.

Reviewer’s comment: In terms of FTIR analysis, where is Fig. S1 or S2 (also NMR results)? I do not have access to the supporting information. It seems its missing. Moreover, for the chemical bonding position of the samples, please cite the reference paper or book.

Authors response: We have corrected and double checked all the Figure numbers and also updated in the supporting information as well. For the chemical bonding position of the samples, appropriate references has also been added in FTIR and 1HMNR studies. Page 12 reference # 17 & page 13 reference # 31.

Reviewer’s comment: In terms of solubility table, following the data is very hard due to the high number of samples. Please re-organize the data to make them much easier to understand. For example, authors can provide a table for F, T, I, PF, PT, and PI samples and put the concentration of each sample in a single table.  

Authors response: We have made a compressed form of the table to incorporate all the data to occupy minimum space in the paper.

Reviewer’s comment: Quality of Fig. 4 and 5 is very low. Please also label all the X-ray peaks for different samples.

Authors response: Figure 4 and 5 has been modified with high quality. WXRD patterns and SEM images have also been labelled.

 Reviewer’s comment: The EDS results in Figure 6 is not giving us any information at the present format. Authors need to bring the quantitative results in a separate table and explain the results in more-details.

Authors response: We have performed EDX along with SEM images on the same instrument just to confirm the presence of iron and silicon atoms in the structures and found positive results that were displayed in the Figure 6. More text is added.

Minor problems:

Reviewer’s comment: The is a type in the X-ray diffraction part line 3, it should be Figure 4 instead of 4.

Authors response: It has been corrected now.

Round 2

Reviewer 1 Report

The manuscript was significantly improved, but this still requires minor revision.

Fig. 4. WXRD-patterns are still in the different scales. This should be corrected. It is impossible to compare XRD-patterns in different scales.

Author Response

Fig. 4 is updated, as suggested by reviewer 1. 

Reviewer 2 Report

The paper is ready to publish in a present form.

Author Response

We thank reviewer 2.